# Experimental Investigation of Crystal Blocking in Drainage Pipes for Tunnels in the Karst Region

Chongbang Xu [1], Yang Chen [2], Yunxuan Yang [2], Pengfei Li [2,*], Siqing Wang [2] and Lei Li [1]

[1] Research Institute of Highway Ministry of Transport, Beijing 100088, China
[2] The Key Laboratory of Urban Security and Disaster Engineering, Ministry of Education, Beijing University of Technology, Beijing 100124, China
* Correspondence: lpf@bjut.edu.cn

**Abstract:** Crystal blockage of tunnel drainage pipes is one of the main causes of problems such as lining cracking and water leakage. The study of the crystal development rule is of great significance for the design of tunnel drainage systems and the long-term safety of tunnel support structures. In this paper, a series of experimental studies on the crystallization development law of drain pipes are conducted. The effects of the connection method of the drain, the diameter of the pipe, the spacing of the circular drain, and the material of the drain on the crystallization development pattern are investigated. The results show that the groundwater environment has a great influence on the crystallization development of the drain pipe. As the drain diameter and the spacing between two adjacent circular drains increased, the time for complete blockage of the drain is prolonged. The rate of crystallization on the drainage pipe can be effectively reduced by changing the material of the drainage pipe from polyamide (PA) to polypropylene (PP). The present study provides a reference for research work related to crystallization blockage in tunnel drainage pipes.

**Keywords:** model test; drainage pipe; anti-crystallization; crystal blockage





## 1. Introduction

With the substantial increase in tunnel construction, the problems of tunnel drainage systems have become increasingly prominent, such as tunnel lining cracking and water leakage in side wall, which have seriously affected the use and safety of tunnels [1,2]. Once the drainage system is blocked, the water pressure behind the lining rises, leading to increases in the axial thrust and bending moment of tunnel lining and a severe threat to the long-term safety of tunnels [3–7]. A large expenditure has to be made to monitor and maintain the tunnel drainage system continuously [8,9]. Therefore, it is of great significance to conduct experimental research on the development law of internal crystal blockage of the drainage system to ensure the safety and stability of the tunnel lining structure.

A great amount of effort has been devoted to investigating the blocking problem in drainage pipes for tunnels. Zhang and Zhou conducted a large number of on-site surveys in the field of drainage and found through microscopic analysis that the main component of crystallization is calcite crystal calcium [10,11]. It is proposed that the continuous deposition and blockage of calcium carbonate minerals such as calcite ($CaCO_3$) will seriously affect the water transport in tunnel water channels and pipeline systems [12]. It is revealed by a study of old tunnels in South Korea that the sediments in the drainage system of tunnels are mainly due to the chemical reaction between the tunnel concrete lining and groundwater. Yoon analyzed the scale sample of tunnel drainage pipe, and pointed out that calcite crystal is the main component of the scaling sample of tunnel drainage pipes [13].

The substance and formation mechanism of the blockage of the tunnel drainage system is another issue which concerns the researchers most. A large number of experiments have found that square carbonate is the main substance of the blockage of the tunnel drainage

system. Some scholars analyzed the influencing factors of the formation mechanism of carbonate. Larsen conducted a series of laboratory experiments to obtain the scaling mechanism of calcium carbonate based on the crystal growth kinetic theory [14,15]. Xu and Li found that the release of $CO_2$ gas caused the pH of the solution to rise, thereby promoting the precipitation of $CaCO_3$ [16,17]. The precipitation of $CaCO_3$ changed the ratio of the total carbon concentration in the solution to the total calcium concentration, leading to a change in the $CO_2$ precipitation rate. Xing found that the increase in pH increases the thermal resistance of progressive fouling. However, the flow rate plays a more critical role in the thermal resistance of progressive fouling compared with pH [18]. Jung found that installing a magnetization device and quantum stick in the tunnel drainage pipe can effectively reduce the formation of blockages and that the quantum stick shows a reasonable descaling effect under conditions that exceed the speed limit [19]. According to the physicochemical mechanism of $CaCO_3$ precipitation, the fluid–solid interaction of highly soluble concrete components, silicate, for instance, enhances the exchange of $CO_2$ between the drainage solution and tunnel atmosphere [20,21]. Zhang et al concluded that the experimental conditions such as the solution chemistry, the flow rate, the temperature, and the saturation index may have significant impacts on $CaCO_3$ deposition kinetics. They found that flocked polyvinyl chloride (PVC) pipes have a certain effect on the removal of crystallization of tunnel drainage pipes [22–24]. Chen found that calcite depends not only on the hydraulic conductivity and $CaCO_3$ content of surrounding rocks but also on the lining materials and the geometry of tunnels [12]. Liu found that the flocking drainage pipe has an anti-crystallization effect, and the efficiency of the flocking drainage pipe can increase by more than 50% compared with that of ordinary drainage pipes [25–28].

The aforementioned literature research recognized that the main substance of the crystallization blockage of tunnel drainage pipes is carbonate. However, there has been little research on the crystallization development law of circular drainage pipes and longitudinal drainage pipes in combination with the layout characteristics of tunnel drainage pipes.

Therefore, this work presents a series of experiment-based studies of the drainage system subjected to crystal blockage under the actual tunnel drainage system layout environment. In order to study the law of crystal clogging in tunnel drainage pipes, an experimental equipment installation consisting of a drainage system, a water supply system, and a test chamber is established. Consequently, the influences of the connection method of the drainage pipe, the pipe diameter, the distance between the circular drainage pipes, and the material of the drainage pipe on the crystallization law are investigated. (The factors that have less influence on the blockage of the tunnel drain crystal are not considered in this paper). It is found in the present work that the main blocking substance of the tunnel drainage system is calcite, and its formation is related to the ion concentration of the groundwater, the height of the groundwater level, and the temperature. Some useful suggestions are also proposed for preliminary design purposes.

## 2. Experimental Equipment

Crystal blockage of tunnel drainage pipelines is an important cause of lining cracking and leakage. In order to study the crystallization plugging law of drainage pipes, this paper conducts the plugging test of drainage systems in the laboratory. The model test chamber consists of three parts, which are composed of the water supply system, the drainage system, and the test chamber. The water supply system allows the test solution to circulate in the device by controlling the pressure and the size of the flow rate of water; the test chamber and the drainage system simulate the whole process from blockage to dissolution of crystals in the whole tunnel, and then study the change law of crystal crystallization in the drainage pipe. The system can also study the influence of the type of geotextile, the material of the drainage pipe, and the type of blockage on the blockage pattern of the tunnel geotextile as well as on the overall drainage performance of the tunnel. As shown in Figure 1a, detailed presentations of the three components are shown as follows.

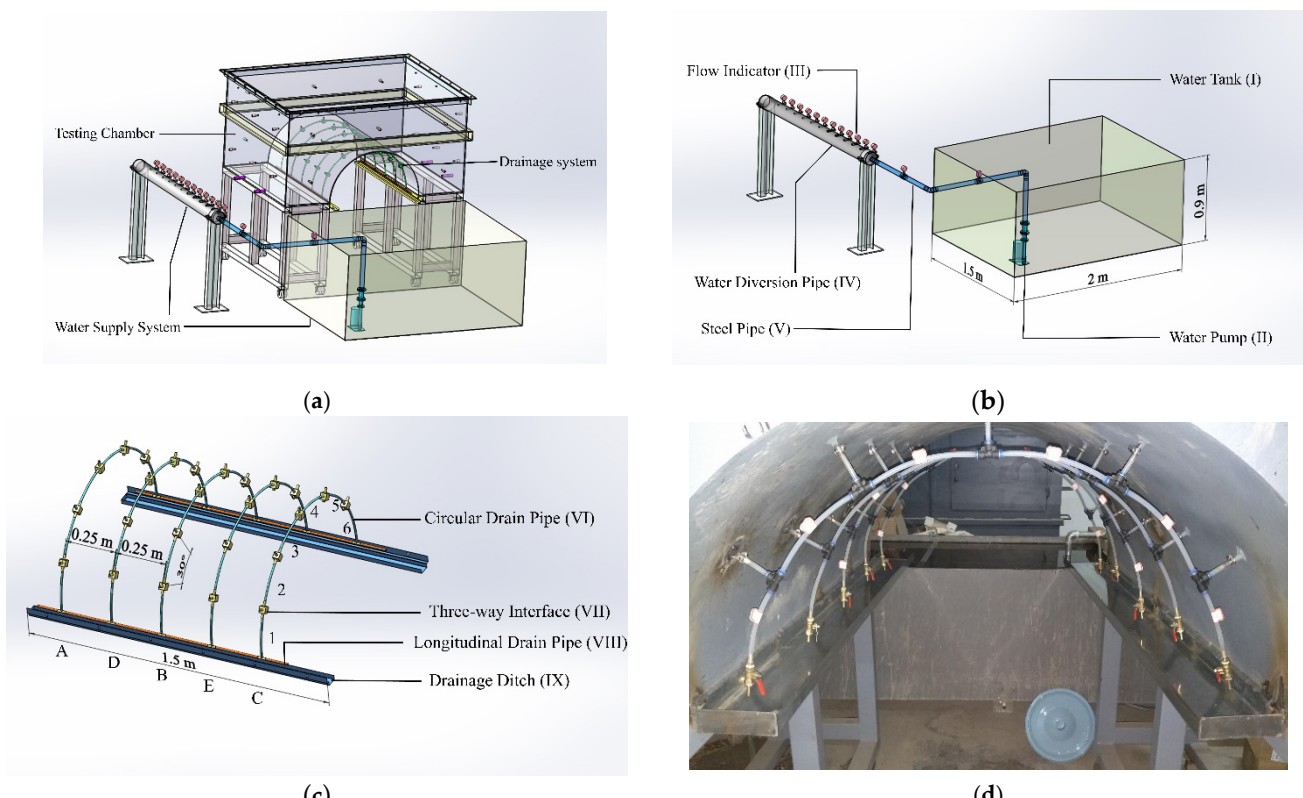

**Figure 1.** Sketch of experimental equipment for drainage system. (**a**) The overall equipment. (**b**) Schematic diagram of water supply system. (**c**) Schematic diagram of the drainage pipes. (**d**) Physical equipment of the drainage pipes.

### 2.1. Water Supply System

The water supply system consists of five components, as shown in Figure 1b, which provide the power for water flow and receive water from the drainage system. The stainless-steel water tank (I) with the dimensions of 2.0 m × 1.5 m × 0.9 m (length × width × height) is used to hold ionic solution. The water pump (II) is used to transport the solution to the water distributor through the steel pipe (V). The water diversion pipe (IV) is connected to the water intake pipe with hoses, shunting the solution to the testing chamber. The flow indicator (III) is set at the outlet of the water diversion pipe to observe the water pressure at different inlets.

### 2.2. Drainage System

The drainage system, which is composed of 6 mm-diameter-polyamide circular drainage pipes (VI), three-way interfaces (VII), 8 mm-diameter-polyamide longitudinal drainage pipes (VIII), and drainage ditches (IX), is set at the bottom of the testing chamber as shown in Figure 1c,d. The circular drainage pipes are connected to the two longitudinal drainage pipes with a slope of 1% at 25 cm intervals. The side ditches with a slope of 3% are laid along the longitudinal direction of tunnels, outside of the bottom of the test box. The circular flow path of the solution from the water tank (I) to the drainage ditches (IX) is shown in Figure 2.

### 2.3. Testing Chamber

The schematic and physical models of the test chamber simulating the upper part of the tunnel structure and the groundwater environment are shown in Figure 3a,b, respectively. The dimensions of the testing chamber are 2.00 m × 1.50 m × 1.00 m (length × width × height).

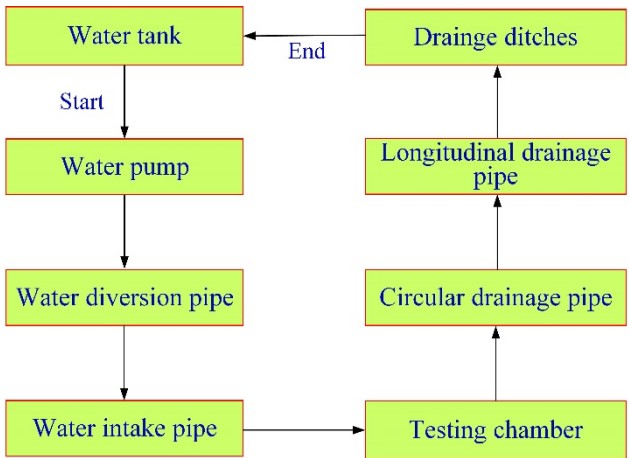

**Figure 2.** Schematic diagram of solution circulation.

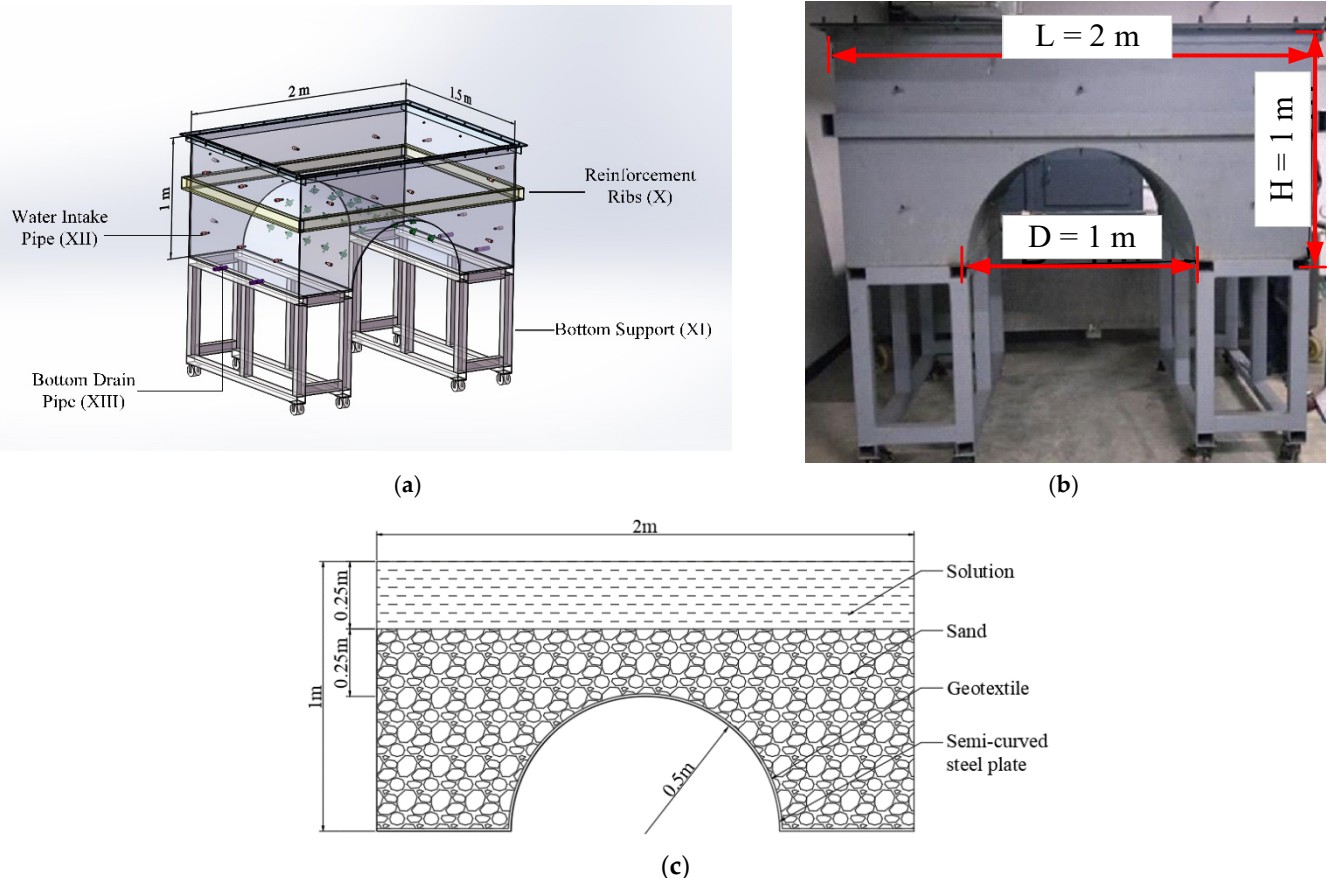

**Figure 3.** The testing chamber and the cross-section of tunnel. (**a**) Schematic diagram of the testing chamber. (**b**) Physical photograph of the testing chamber. (**c**) Sketch of the cross-section of tunnel.

A semi-circular arc steel plate with a radius of 0.50 m is set at the bottom of the testing chamber to simulate the initial support of tunnels. Drainage pipes in the model test are equidistantly distributed in the longitudinal direction with 5 sections, and each section has 5 holes which are separated by 30°. The 5 sections are named A, B, C, D, and E, respectively. Six drainage pipes are installed in each section, numbered 1, 2, 3, 4, 5, and 6 respectively. Therefore, each pipe can be identified by its section name and pipe number, e.g., C1, B2, and D6. 28 water intake pipes (XII) on the outside of the testing chamber are divided into two rows with a distance of 0.50 m. Two bottom drain pipes (XIII) with a diameter of

10.0 mm for discharging the remaining solution are set on both sides of the bottom of the testing chamber. A one-meter-high stainless-steel bracket is arranged at the bottom (XI).

Stainless reinforcement ribs (X) of 10.0 mm thickness and a hoop are welded at the surroundings and a height of 0.50 m above the testing chamber, respectively, to prevent excessive deformation. Besides, the top cover is connected to the testing chamber with a sealing strip to prevent overflowing of the solution.

As illustrated in Figure 3c, the tunnel surroundings are simulated by a sand layer, while geotextiles and semicircular steel plates are installed to simulate the initial support and lining of the tunnel respectively, thus enabling a more realistic simulation of the tunnel groundwater environment.

## 3. Testing Design

### 3.1. Ionic Solution Preparation

In the present study, the anhydrous calcium chloride ($CaCl_2$) and sodium bicarbonate ($NaHCO_3$) are used to simulate groundwater ion crystallization. Test ion concentrations are: $Ca^{2+}$: 9 mmol/L and $HCO_3^-$: 8 mmol/L. The mechanism of $CaCO_3$ scaling is based on its precipitation–dissolution balance as follows:

$$Ca^{2+} + 2HCO_3^- \Leftrightarrow CaCO_{3+}CO_2 + H_2O \tag{1}$$

In this study, it is assumed that the ion recharge source is continuous since the ions in the groundwater are mainly from the surrounding rock and atmosphere. The device is tested every 12 h and the corresponding missing material is replenished according to the changes in the ion concentration detected in the collection tank.

### 3.2. Formation Parameters

The test fill particles are taken from the foundation pit of a subway station in Beijing PRC, consisting of more than 90% sand and less than 5% clay particles, and an average particle size of about 0.17 mm, as shown in Figure 4. The measured maximum and minimum dry densities of the sand are 1950 kg/m$^3$ and 1400 kg/m$^3$, respectively. The water content, cohesion, and internal friction angle are 6.54%, 0.30 kPa, and 35°, respectively. Besides, the elastic modulus, Poisson's ratio, and permeability coefficient are 35.0 MPa, 0.3, and $1.1 \times 10^{-3}$ cm/s, respectively.

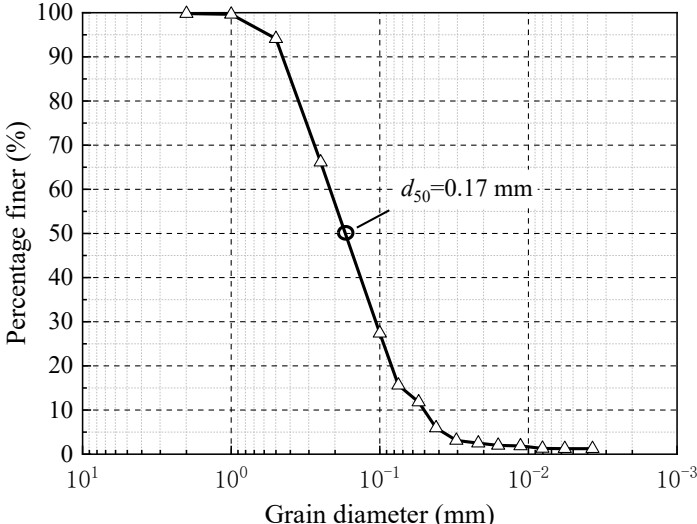

**Figure 4.** Gradation curve of sand particles.

### 3.3. Experiment Procedure

Since the temperature of the project site is difficult to control, in order to ensure the accuracy of the experiment, the experiment was carried out at standard room temperature. The presented test mainly focuses on the stage carried out in the laboratory, and the detailed experimental steps are shown in Figure 5. Specific steps are as follows.

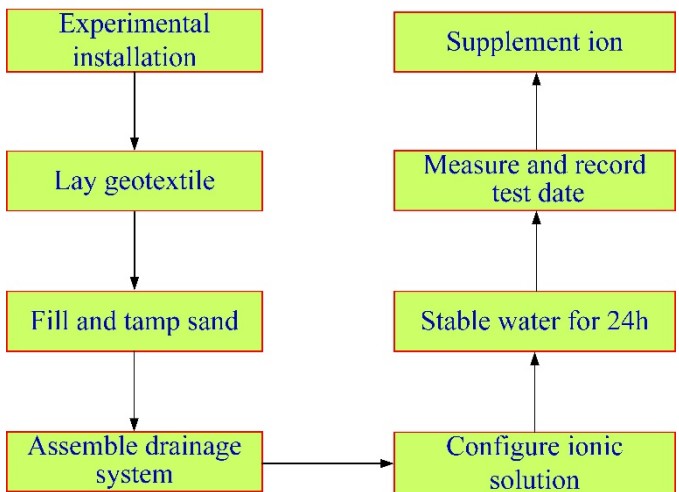

**Figure 5.** Schematic diagram of experimental steps.

Firstly, fix the test box, water collection tank, water separator, and other large equipment in preparation for the test site, connect the test equipment, and check whether any instruments leak. Lay the non-woven fabric on the inside bottom of the test box and seal the edges with waterproof glue to prevent the loss of small-sized sand and gravel. In addition, fill the soil inside the test box to a height of about 70 cm, tamped and leveled.

Secondly, cut the drainage pipes and number them sequentially. Connect the numbered circular drains to the bottom of the test box in sequence as shown in Figure 3a. Place the bucket on the electronic scale, pour in the water, record the water volume, calculate the required mass of calcium chloride ($CaCl_2$) and sodium bicarbonate ($NaHCO_3$) according to the ionic concentration, weigh out the corresponding mass of the mother liquor, pour it into the bucket evenly, and stir slowly to dissolve it fully. The prepared ionic solution is injected into the water collection tank and this operation is repeated until the requirements are met.

Thirdly, the experimental data and phenomena were recorded. When the air inside the test chamber is completely exhausted, close the drain valve, then open the drain valve in the tunnel and monitor the water pressure value of the water injection hole, and keep the leakage stable when it reaches 50 kPa. Use a measuring cylinder to collect the water discharged from the drainage port, and record the time required to collect 50 mL of water. Turn off the water pump of the water collection tank, open the water valve at the bottom of the test chamber, and wait until the groundwater in the test chamber is completely discharged. Remove the drainage pipe, measure the weight of each water pipe after drying and record it, and install the water pipe to its original position after the re-recording is completed. Then, test the ion concentration of groundwater in the water collection tank every 12 h and add the appropriate substance according to the change in ion concentration.

Finally, repeat the above operation until the crystal weight is stable. Then, complete each set of tests and record the data according to the predetermined working conditions.

### 3.4. Experimental Conditions

In this experiment, the effect of longitudinal drainage pipes (LDP) with different diameters, spacing, and materials on the development pattern of crystalline blockage is investigated with and without fillers. The test conditions are shown in Table 1.

**Table 1.** Indoor test conditions of tunnel drainage pipe blockage.

| Condition | Sand | LDP | Diameter | Spacing | Material |
|---|---|---|---|---|---|
| Standard condition | Yes | No | 6 mm | 50 cm | PA |
| No filling sand | No | No | 6 mm | 50 cm | PA |
| Install LDP | Yes | Yes | 6 mm | 50 cm | PA |
| Enlarge pipe diameter | No | No | 8 mm | 50 cm | PA |
| Narrow section space | No | No | 6 mm | 25 cm | PA |
| Change pipe material | No | No | 6 mm | 50 cm | PP |

Note: PA denotes polyamide, and PP denotes polypropylene.

Five parameters are considered in this experiment. The first one is filled with sand. The unfilled sand condition is designed to contrast with the filled sand condition. For the no-filled sand condition, the test chamber is filled with the ionic solution. Clearly, the flow rate in the drain for the no-fill sand condition is greater than that for the sand-fill condition. Therefore, this comparison can reflect the effect of flow rate on crystal blockage. The second influencing factor is the longitudinal drainage pipe (LDP). Two longitudinal drainage pipes are installed at the ends of the circular drainage pipes, respectively. They are used to collect the water from the circular drainage pipes. Obviously, there are more connections if the longitudinal drainage pipe (LDP) is installed. Therefore, this comparison could reflect the influence of connection conditions on the crystal blocking.

It is believed that the pipe diameter had a significant effect on the crystal clogging law, so the pipe diameter is selected as the third influencing factor to be studied. Therefore, on the basis of previous scholars' research, a drainage pipe with a diameter of 8 mm is selected for comparison experiments to study the analysis of the effect of pipe diameter on the crystal clogging law.

The section space of two adjacent drainage pipes is another important factor which determines the crystalline clogging law of drainage pipes. Obviously, the total number of circular drainage pipes will increase with the decrease in the spacing of the circular drainage pipes, which can improve the drainage efficiency of the drainage system. Therefore, the distance between the original two circular drainage pipes is considered to be reduced by half and used to study the effect of spacing change on the crystallization clogging law. Finally, there is no doubt that the pipe material can also have a significant impact on the crystal clogging law. Based on the research of previous studies, it is found that polypropylene (PP) and polyamide (PA) have similar properties. Therefore, in this paper, polypropylene (PP) is used instead of polyamide (PA) for comparative analysis to investigate the effect of drainage pipe materials on the development law of crystalline blockage.

## 4. Results and Analysis

### 4.1. Standard Condition

Crystal obstructions are formed and accumulated on the inner wall of the drainage pipes and the drainage side ditch, causing blockage of the drainage system after 12 days of testing, as shown in Figure 6a,b, respectively. The yellow crystal obstructions are mainly $CaCO_3$ and sediment, as shown in Figure 6c.

The crystallization curves of the three circular drainage pipes (A, B, and C) in the standard condition are shown in Figure 7a. The curves illustrate that the development rule of crystals on the inner wall of the drainage pipe under working conditions can be mainly divided into two stages: (I) the growth stage and (II) the stabilization stage. In regard to the first stage (0–9 d), with the continuous flow of the ionic solution, the crystals formed inner side of the drainage pipe quickly for the rate of ion formation is greater than the rate of solution scouring the inner wall, leading to a continued increase in the weight of drainage pipe. With regard to the second stage (10–12 d): After 9 days, as the crystal formation rate gradually equals the solution scouring rate, the growth rate of crystals slowed down and the crystal weight reached the peak (A:1.95 g/m, B:1.82 g/m, C:1.94 g/m) at the 12th day, resulting in a stable amount of crystallization.

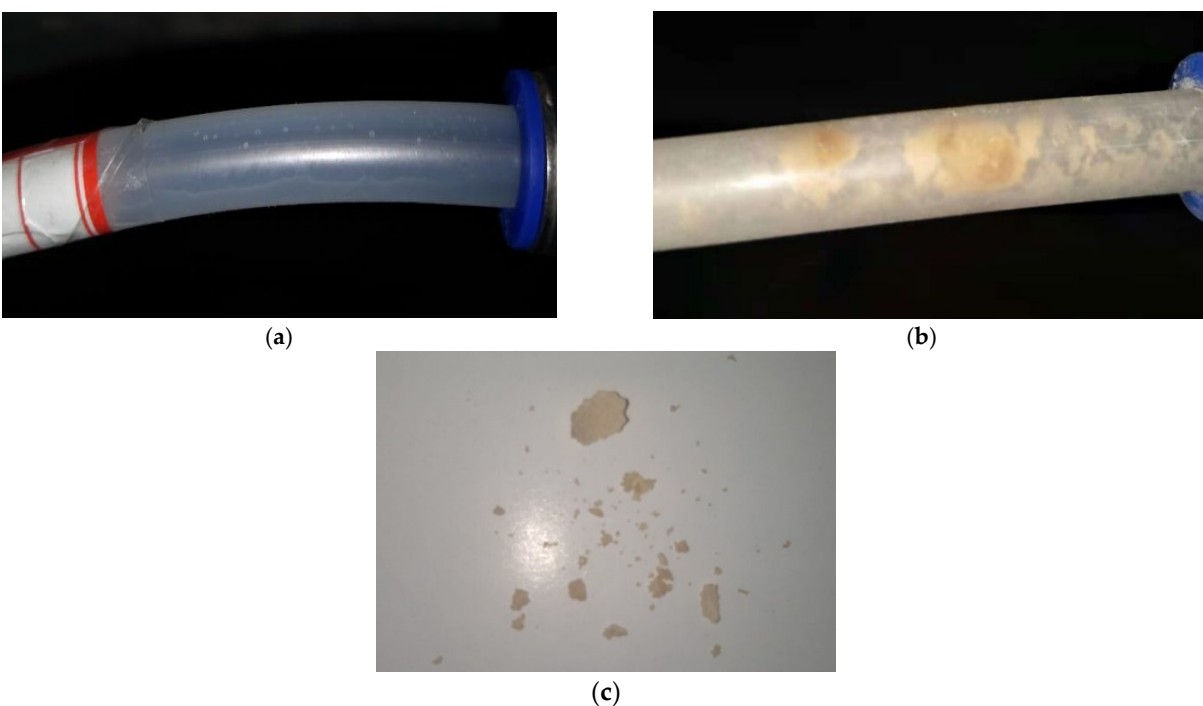

**Figure 6.** Drainage pipes before and after the testing and the crystal sample. (**a**) Drainage pipes before the testing. (**b**) Drainage pipes after the testing. (**c**) The crystal sample.

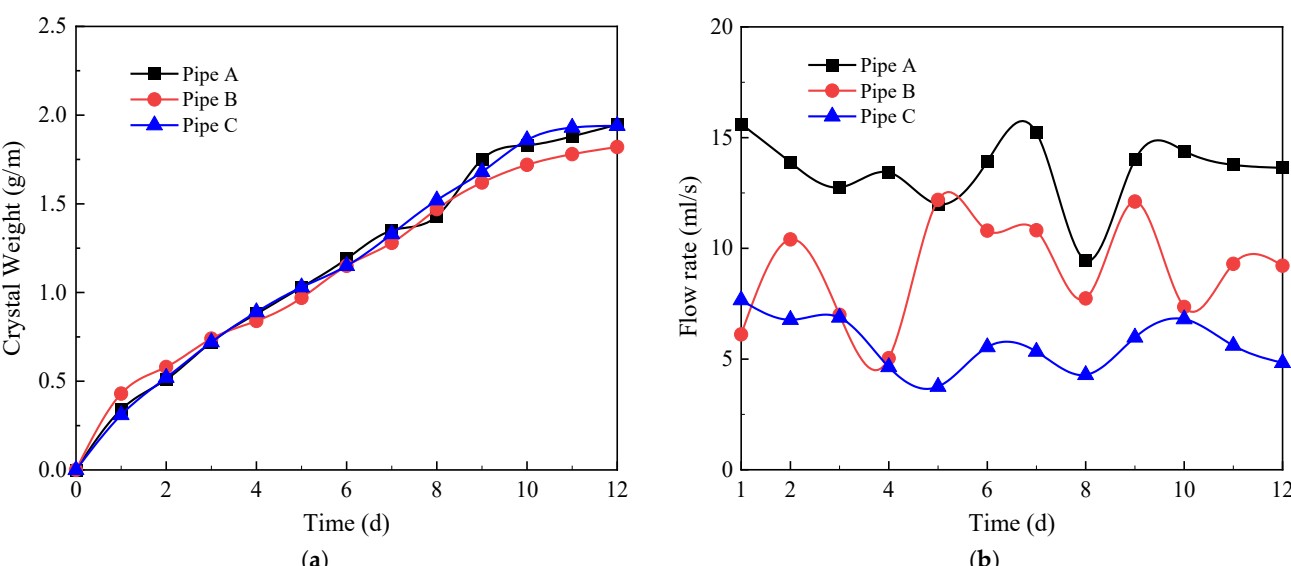

**Figure 7.** Variation of crystallization rate and flow rate of different circular drainage pipes with time. (**a**) Crystallization rate curves. (**b**) Flow rate curves.

The variation curves of the three circular drainage pipes (A, B, and C) versus time in the experiment are shown in Figure 7b. The flow curve of the ABC pipeline fluctuates greatly, and there are many intersections, which cause the instantaneous crystallization rate and the scouring rate of each pipeline to be similar, so the crystallization curve of each circular drainage pipeline is basically consistent.

An illustration of the crystallization of the arch (tubes 1 and 6 as shown in Figure 1c, spandrel (tubes 2 and 5 as shown in Figure 1c) and vault (tubes 3 and 4 as shown in Figure 1c) versus time under standard conditions is shown in Figure 8. It is illustrated from the crystallization time curve that the crystal weights are 1.93 g/m, 1.87 g/m, and

1.44 g/m for arch, shoulder, and vault, respectively, due to the following two reasons. Firstly, the sequence of the locations of flow rates is arch > spandrel > vault. When the flow rate is not enough to scour the tunnel crystal, with the increase in flow velocity, more ions pass through the drainage tube in unit time, leading to an increase in the amount of crystallization. Secondly, the unstable crystals formed on the upper part of the circular drainage pipe are vulnerable to being peeled off under the erosion of the water flow, causing the crystals at the bottom to increase. Therefore, the test results show that the bottom blockage should be given priority and prevented when considering the long-term operation of annular drainage pipelines.

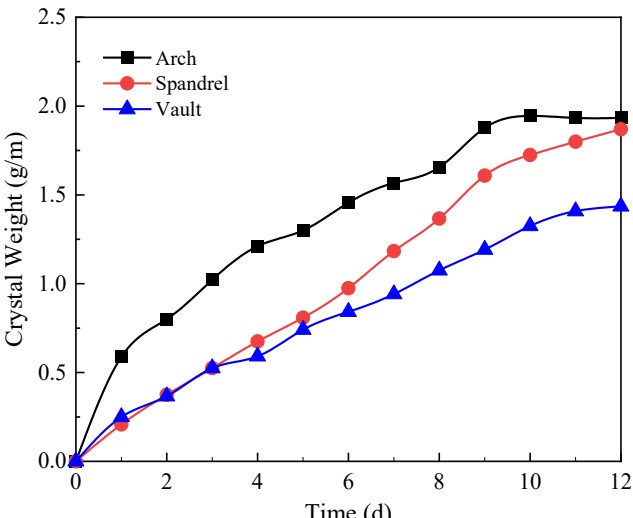

**Figure 8.** Crystal weight change curves at different parts of circular drainage pipes.

### 4.2. Influence of Overlaying Soil

The time history curves of the crystallization on the drainage pipes both with and without the presence of overlaying soil are shown in Figure 9a. It can be seen that the crystals increase rapidly (0–4 d) and the crystallization rate of the no-filling condition is significantly greater than that of the filling condition. The maximum difference is that the soilless working crystallization curve has a descending section (5–6 d). This is maybe because the crystals initially formed in the inner wall of the drainage pipe are not stable at high flow rates, and continue to fall off under the erosion of the water flow. The crystallization rate of the inner wall of the drainage pipe is less than the scouring rate of the solution, resulting in a decrease in crystallization. The crystals formed inside the drainage pipe tend to be stable, and their adhesion to the inner wall is stronger than the scouring force of the solution. The rate of crystal formation at this stage is greater than the scouring rate of the solution, and the crystallization continues to increase (6–8 d).

The time history curve of the flow rate for each drainage pipe is shown in Figure 9b. The average unit flow rate of a single drainage pipe under the soilless condition is seven times larger than the filled condition due to the buffering effect of sand. The initial crystallization rate increases with the increase in ions that pass through per unit time, indicating that the water flow rate is an important factor affecting the formation of crystals. At the beginning of the test, the crystallization rate of the blockage is stronger than the scouring rate of the water, which led to the continuous deposition of the blockage and the increase in the crystallization amount, and the drainage performance decreased continuously. When the scouring rate and crystallization rate reached equilibrium, the blockage no longer increased, and the tunnel drainage performance remained stable.

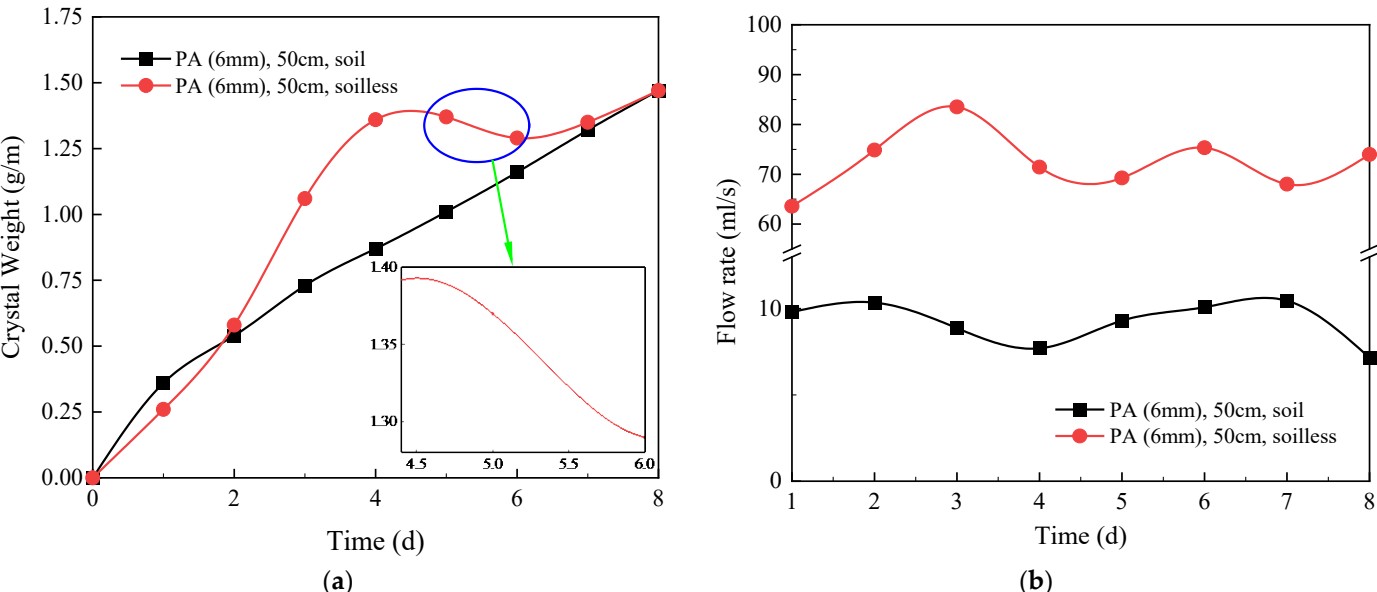

**Figure 9.** Variation of crystallization rate and flow rate of circular drainage pipes with time. (**a**) Crystallization rate curves. (**b**) Flow rate curves.

The time history curve of the crystallization on the circular drainage pipes is shown in Figure 10. The crystallization started to decrease rapidly with the ratio of 26.2%, 10.0%, and 6.9%, respectively, after the crystallization amount of each part reached the peak on the fourth day (arch: 1.6 g/m; spandrel: 1.3 g/m; vault: 1.2 g/m). At this time, due to the higher flow rate in the lower part of the annular drainage pipe, the crystallization stripping rate is much larger than the crystallization formation rate, resulting in the reduction of crystallization in the lower part. With the gradual accumulation of crystals, the scouring rate of the solution is smaller than the crystallization rate, and the crystals again show a rising trend.

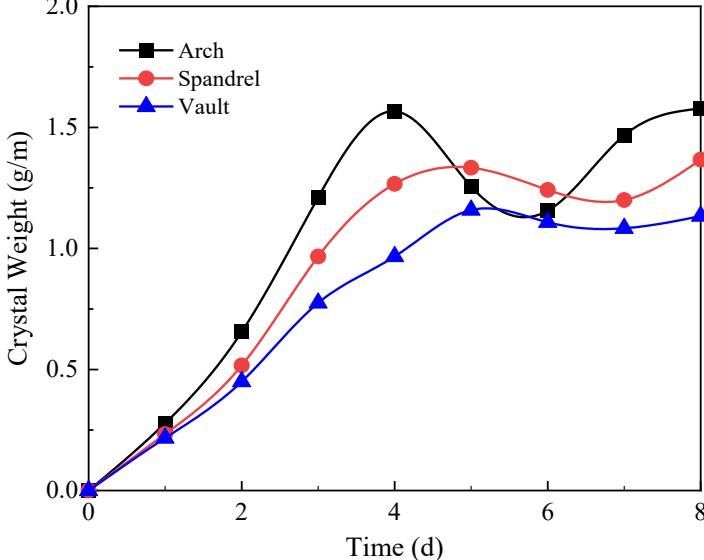

**Figure 10.** Crystal weight variation curve of circular drainage pipes.

### 4.3. Influence of the Longitudinal Drainage Pipe

As shown in Figure 11, on the 5th day of observation, sediment flowed into the longitudinal drain and began to accumulate, resulting in an increase in the number of crystals in the longitudinal drain. The reason for this phenomenon may be that the slope of the longitudinal drain is low, making it easier for sediment to be deposited in the drainage pipe. Figure 12a shows the crystallization curves of the tunnel drainage pipe versus time after installing the longitudinal drainage pipe. In the first 6 days, the crystal weight at the inner wall of the longitudinal drainage pipe increases rapidly and reaches a short-term peak of 2.67 g/m. On the 7th day, the amount of crystallization decreased by 6%, while continuing to grow to the maximum value of 3.10 g/m from the 8th day to the 12th day. It may be because the unsteady crystals formed are washed away by the large flow of the longitudinal drainage pipe, resulting in a short-term decrease in the amount of crystals. After that, the crystallization rate is greater than the scouring rate, and the crystallization continues to grow steadily.

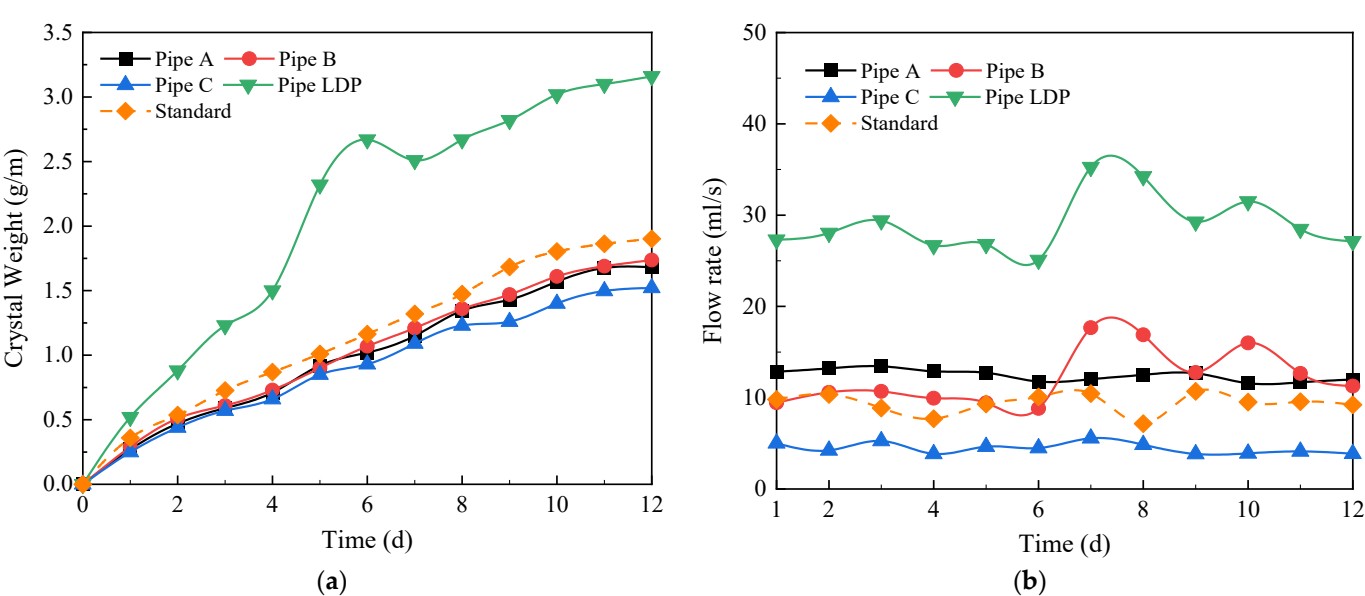

**Figure 11.** Photo of sediment deposition.

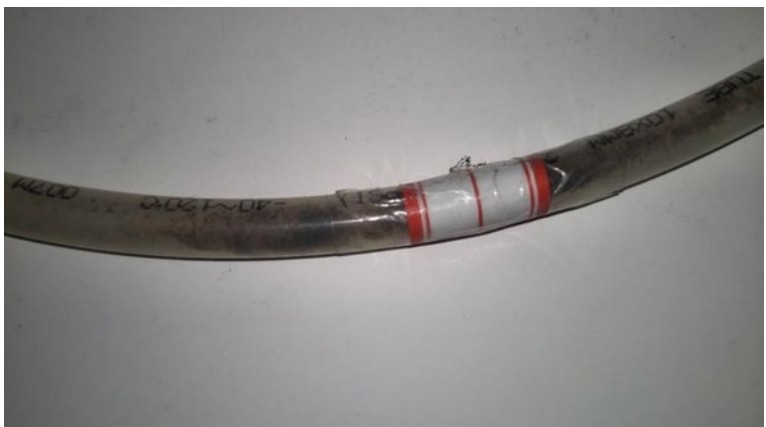

**Figure 12.** Variation of crystallization rate and flow rates of drainage pipes with time under different conditions. (**a**) Crystallization rate curves. (**b**) Flow rate curves.

The amount of crystallization in the longitudinal drainage pipe is about twice greater than that of the circular drainage pipe (1.50 g/m,1.69 g/m, and 1.68 g/m, respectively, corresponding to A, B, and C) due to the following reasons: Firstly, all the water flowing

out of the circular drainage pipe flows into the longitudinal drainage pipe, resulting in a large flow of the longitudinal drainage pipe and a large amount of ions passing through the unit time, as shown in Figure 12b; secondly, due to the effect of circulating water flow, a large amount of crystals falling from the circular drainage pipe flow into the longitudinal drainage pipe; thirdly, the slope of the longitudinal drainage pipe is small compared with the friction force of the crystal body, making the crystal formed at the circular drainage pipe falling off and the inner wall of the longitudinal drainage pipe easily, resulting in a constant amount of crystal.

The test proved that the longitudinal drainage pipe is easier to form crystals and sediment than the circumferential drainage pipe. However, after the installation of the longitudinal drainage pipes, the crystallization amount and flow of the circular drainage pipes basically did not change significantly compared with the standard conditions, indicating that the addition of the longitudinal drainage pipes has no significant effect on the blocking performance of the circular drainage pipes. Therefore, when considering the long-term operation of the tunnel drainage system, priority should be given to the clogging of the longitudinal drainage pipe.

### 4.4. Influence of the Diameter of the Circular Drainage Pipe

After replacing the circular drainage pipe from a 6 mm-inner-diameter PA pipe to an 8 mm-inner-diameter pipe, it is found that all parts of the drain pipe are in a non-full flow state and the drainage system did not fully exert its drainage capacity.

The No. 3 pipe at section C is partially blocked during the experiment, resulting in the flow of the C pipe (3.84 mL/s) being significantly lower than that of the other pipes (8.68 mL/s, 10.53 mL/s, and 9.39 mL/s for pipes A, B, and D, respectively), as shown in Figure 13a. In Figure 13b, the crystal change curves of A and B tubes are basically the same as those of the standard operating conditions, which proves that an increase in the inner diameter of the circular drain pipe barely influences the formation of the crystal amount. However, the final crystal amount of the C tube is significantly lower than those in the other pipes. This may be due to the reason that the flow rate of the C tube is small, and the crystal ions passing through tube C per unit time is also small, resulting in a small amount of crystal.

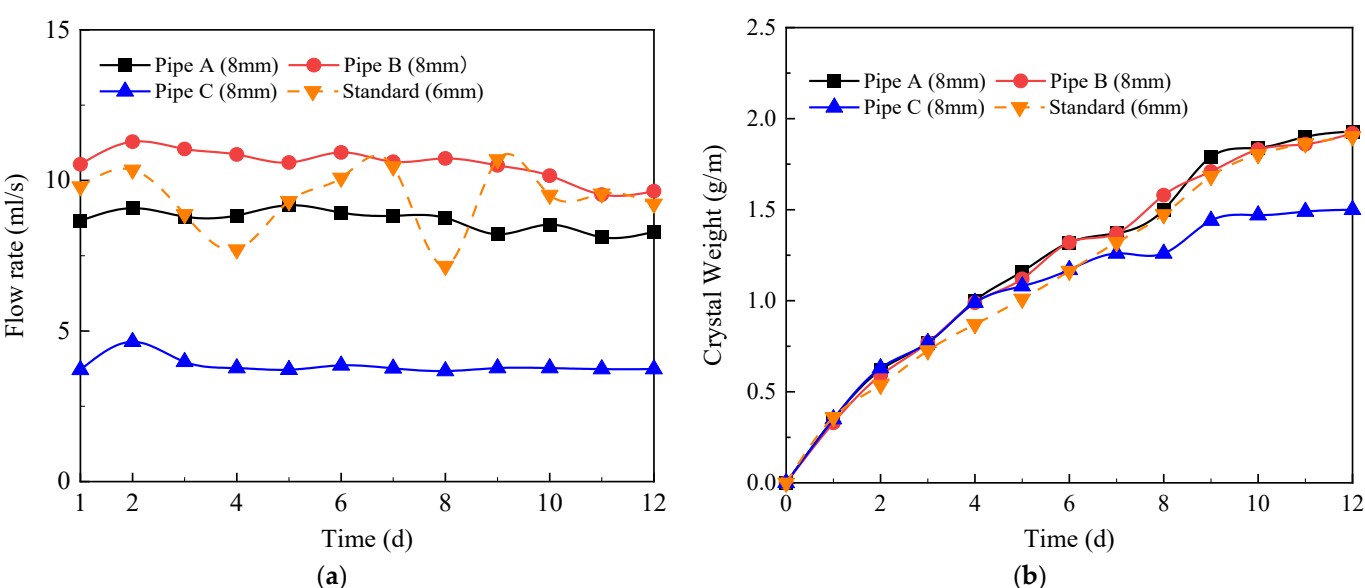

**Figure 13.** Variation of flow rates and crystallization rates of circular drainage pipes at different diameters. (**a**) Flow rate curves. (**b**) Crystallization rate curves.

### 4.5. Influence of the Spacing of the Circular Drainage Pipe

After adjusting the spacing of the circular drainage pipes from 50.0 cm to 25.0 cm, the crystallization rate and flow curves of the drainage pipes are shown in Figure 14a,b, respectively. When the spacing of the circular drainage pipe decreases from 50 cm to 25 cm, the average crystallization of the pipe decreases from 1.90 g/m to 1.75 g/m. This shows that a decrease in spacing will lead to a decrease in the crystallization amount of each circular drainage pipe, but the overall crystallization amount still shows an upward trend, and the growth trend is basically consistent.

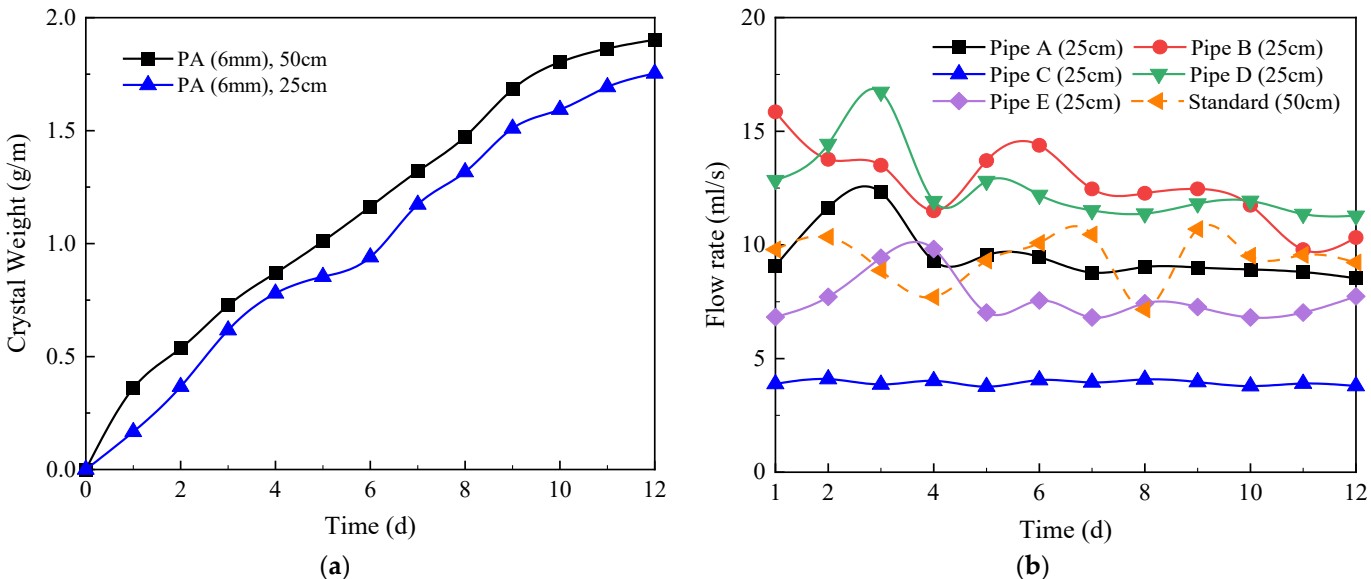

**Figure 14.** Variation of average crystallization rate and flow rates of circular drainage pipes under different spacing. (**a**) Average crystallization rate curves. (**b**) Flow rate curves.

After reducing the spacing of the circular drainage pipes, the flow rate of the circular drainage pipes (A, B, C, D, and E) and the standard condition (F) remained in the same order of magnitude. However, there is a 65% increase in the drainage efficiency when the total drainage volume increased from 28.18 mL/s of the standard working condition to 46.58 mL/s. Thus, reducing drainage pipe spacing can effectively improve drainage capacity. For tunnels with large drainage capacity, it is better to reduce the spacing of the circular drainage pipes to improve the overall drainage performance.

### 4.6. Influence of the Material of the Circular Drainage Pipe

The crystallization change curve of the circular drain pipe after replacing the PA pipe with the PP pipe is shown in Figure 15a. During the test, the crystallization rates of the PP pipe and PA pipe are 0.01–0.02 g and 0.2–0.3 g per day, respectively, indicating that the inner wall of a PP pipe is less susceptible to crystallization, compared with a PA pipe. The PP pipe is helpful to prevent the blocking of the drainage pipe and can effectively reduce the formation of crystals. However, the PP pipes age easier and are of low strength, so the selection of materials should be based on the actual situation of the project.

As shown in Figure 15b, after replacing the PA pipe with a PP pipe, the flow rate of the circular drainage pipes (A, B, and C) of the PP pipe is similar to the average flow rate of the PA pipe under the standard working condition. In the first three days, the flow velocity showed a decreasing trend, and then fluctuated up and down within a certain range, with a small variation range. It shows that changing the material of the circular drainage pipe has little effect on the drainage efficiency of the drainage system. Therefore, the material of the drainage pipe is not the main factor affecting the drainage system.

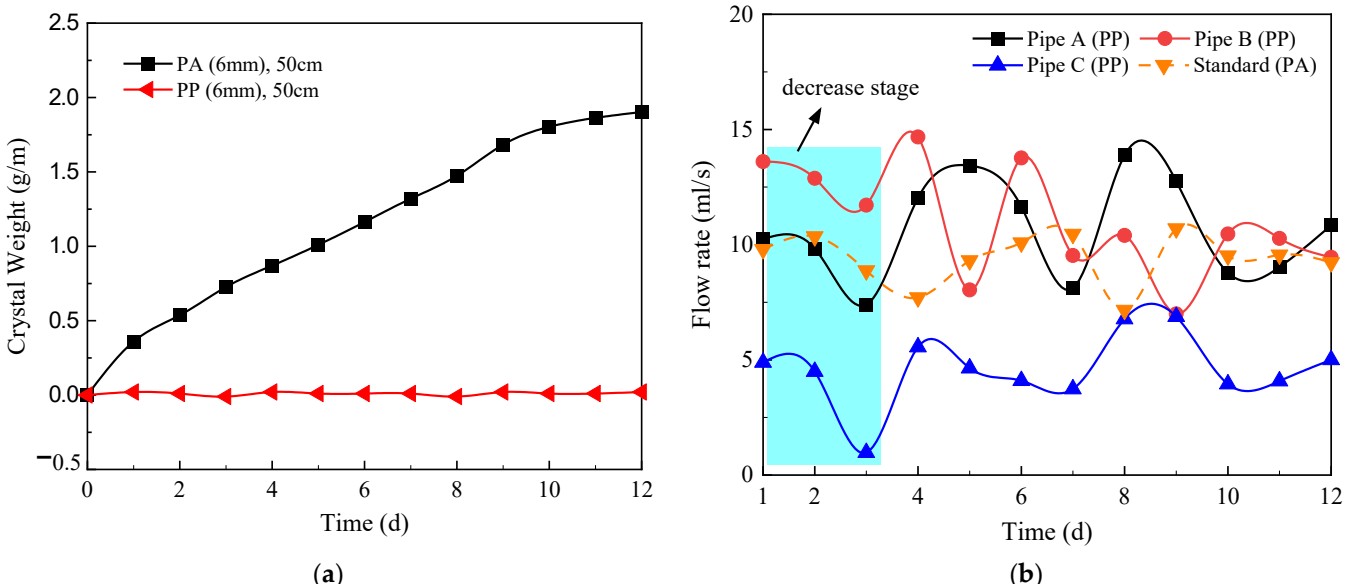

**Figure 15.** Variation of crystallization rate and flow rates with time of drainage pipes of different materials. (**a**) Crystallization rate curves. (**b**) Flow rate curves.

## 5. Conclusions

In the present work, an experimental study is conducted to investigate the development law of crystalline blockage of tunnel drainage pipes. The experimental equipment composed of a testing chamber, drainage system, and water supply system is built, and the crystallization clogging tests are conducted. The effects of five influencing factors, such as the diameter of circular drainage pipe, spacing, material, longitudinal drainage pipe, and the presence or absence of overburden, on the development law of crystallization are studied. Some suggestions are proposed for preliminary design purposes. The conclusions drawn from this study are as follows:

(1) In the presence of the overburden layer, the development pattern of crystallization of drainage pipes can be divided into a growth stage and a stabilization stage. First, the crystallization showed a linear growth trend during the first 9 days. Subsequently, when the crystallization rate of the pipe wall is approximately equal to the scouring rate of the water flow on the inner wall of the pipe, the crystallization amount reaches a peak state, and the drainage performance of the tunnel remains stable. The flow rate of water is an important factor affecting crystallization.

(2) The sequence of the locations where crystals are generated in descending order is the arch, the shoulder, and the vault. In the design of circular drainage pipes, priority should be given to the degree of blockage in the arch. The crystallization process without filling soil can be divided into the growth stage, the decline stage, and the stable stage. The higher the groundwater flow, the higher the crystallization rate. However, under the soil-free condition, the large water flow has a strong scouring effect on the crystallization, and the crystallization amount shows a decreasing trend in the short term.

(3) The presence of the longitudinal drainage pipes has no significant effect on the blocking performance of the circular drainage pipes. Longitudinal drainage pipes are more likely to form crystals and sediment than circular drains due to high water flow and low slope, causing drain clogging. It is recommended that an appropriate slope of the longitudinal drainage pipes should be selected to reduce crystallization and sediment deposition in actual projects.

(4) The crystallization curve and flow curve are basically consistent with the standard operating conditions when the diameter of the circular drainage pipe is increased from 6.0 mm to 8.0 mm. This indicates that when the drainage system does not reach

100% full flow, the increase in drainage pipe diameter does not have a significant effect on the formation of crystallization and the drainage efficiency of the tunnel. Reducing the spacing between circular drainage pipes can effectively reduce the amount of crystallization of individual drainage pipes and improve the overall drainage efficiency of the tunnel.

(5) The resistance of PP pipes to $CaCO_3$ crystallization is significantly greater than that of PA pipes. In a short period of time, the diameter of the circular drainage pipe and the spacing of the circular drainage pipe have little effect on the resistance to crystallization of the drainage pipes. However, by reducing the spacing of drainage pipes, the drainage efficiency of the tunnel can be significantly improved. It is suggested that reasonable parameters of circular drainage pipes should be selected in the design of drainage systems.

**Author Contributions:** Conceptualization, C.X. and P.L.; methodology, Y.C.; software, Y.Y.; validation, C.X., P.L. and Y.C.; investigation, S.W.; data curation, C.X. and L.L.; writing—original draft preparation, Y.C. and S.W.; writing—review and editing, P.L.; visualization, Y.C.; supervision, P.L. All authors have read and agreed to the published version of the manuscript.

**Funding:** This work was supported by the Major Science and Technology Special Project of Guizhou Province (NO. 20183011). The financial supports are greatly appreciated.

**Institutional Review Board Statement:** Not applicable.

**Informed Consent Statement:** Not applicable.

**Data Availability Statement:** The relevant data are all included in the paper.

**Acknowledgments:** The authors would like to thank Zhicheng Yang and Guohua He for their input on this work.

**Conflicts of Interest:** The authors declare no conflict of interest.

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
