# Peer review of "Experimental Investigation of Crystal Blocking in Drainage Pipes for Tunnels in the Karst Region"

_applsci, doi:10.3390/app122110928_

Round 1

Reviewer 1 Report

The results of a study on blockage of drainage pipes caused by crystallization of calcite are presented. Particularly valuable achievements are the detailed description of the test stand including the testing chamber, drainage system model and ionic solution supply system for the testing chamber, as well as the discussion of the test procedure used. The presented research results made it possible to formulate convincing conclusions about the studied model of the tunnel drainage system. Have the authors considered using any model similarity criteria to transfer to the real object the results of tests performed on the model of the tunnel drainage system?

Detailed remark:

Line 175. What is meant by the term "gold reservoir"? Such a term does not appear either in the drawings of the test stand or in other lines of the manuscript.

Author Response

Thanks very much for your suggestion. The purpose of our research on crystal clogging in drainage pipes is to solve the related problems in engineering through experimental phenomena. Therefore, we did not use the model similarity criterion, but only qualitatively studied the effects of several major factors on causing crystalline blockage to reveal the relevant laws of crystal blockage and to provide a reference basis for engineering sites. In this paper, we further solve the drainage pipe blockage problem in the real project by studying the blockage law of some listed drainage blind pipes. This study provides a reference for the research work related to the crystalline blockage of tunnel drainage pipes. As for Line 175, we want to express the meaning of opening the water tank and the bottom drain of the test tank. Relevant corrections have been made to address inappropriate statements and mis-expressed sentences in this paper. The relevant modifications are as follows:

Open the water tank and the bottom drain of the test chamber. After the air in the test chamber is completely discharged, close the drain valve, then open the drain valve in the tunnel, monitor the water pressure value of the injection hole and keep the leakage stable when it reaches 50 kPa.

Reviewer 2 Report

The paper deals with experimental study of development law of crystalline blockage of tunnels drainage pipe. The effects of five influencing factors are studied. The paper is worth considering for publication into Applied Sciences. However, authors should answer next queries:

1.       Author should clearly emphasized in the paper, are there another effects that have influence on crystalline blockage of tunnels drainage pipe?

2.       Similarity of Laboratory test at room temperature, which is performed in the present study and real structure in the construction site, should be discussed.

Author Response

Thanks very much for your suggestion. There are many causes of drain crystal clogging, and the main influencing factors causing drain crystal clogging are studied in this paper. The influence of the drain connection method, the pipe diameter, the spacing of the circular drain and the material of the drain on the crystallization development pattern is investigated. Other factors that have a smaller degree of influence are not studied in this paper and are described in the paper. We have corrected the inappropriate description. The relevant modifications are as follows:

Therefore, this work presents a series of experiment-based study of the drainage system subjected to crystal blockage under the actual tunnel drainage system layout environment. In order to study the law of crystal clogging in tunnel drainage pipes, an experimental equipment installation consisting of a drainage system, a water supply system and a test chamber is established. Consequently, the influences of the connection method of the drainage pipe, the pipe diameter, the distance between the circular drainage pipes and the material of the drainage pipe on the crystallization law are investigated. (The factors that have less influence on the blockage of the tunnel drain crystal are not considered in this paper). It is found in the present work that the main blocking substance of the tunnel drainage system is calcite, and its formation is related to the ion concentration of the groundwater, the height of the groundwater level and the temperature. Some useful suggestions are also proposed for preliminary design purposes.

Reviewer 3 Report

This paper brings an interesting experimental study on the blockage by calcite in the drainage pipes of a tunnel. The influence of the drainage connection method to pipe, pipe diameter, pipe material and distance between pipes were investigated.

The assembly and experiments were well conducted, from what is seen in this section. The corresponding results were also analyzed by the authors.

However, this reviewer misses information on the instrumentation employed for the tests, such as type, model, range, and precision, and an analysis of their performance in the presence of calcite, during the tests. Besides, a statical analysis of the results is welcome. The experimental procedure (3.3), if clearly written in a text (instead of by numbering steps) seems more convenient to a paper, in this reviewer point of view.

Author Response

Thank you very much for your comments. Information about the experimental setup has been added to the text. As for the experimental procedure (3.3) and other problems and suggestions, we have made relevant explanations and modifications in the paper. The relevant modifications are as follows:

The model test chamber consists of three parts, which are composed of the water supply system, the drainage system and the test chamber. The water supply system allows the test solution to circulate in the device by controlling the pressure and the size of the flow rate of water; the test chamber and the drainage system simulate the whole process from blockage to dissolution of crystals in the whole tunnel, and then study the change law of crystal crystallization in the drainage pipe. The system can also study the influence of the type of geotextile, the material of the drainage pipe and the type of blockage on the blockage pattern of the tunnel geotextile as well as on the overall drainage performance of the tunnel.

Firstly, fix the test box, water collection tank, water separator and other large equipment in preparation for the test site, connect the test equipment, and check whether any instruments leak. Lay non-woven fabric on the inside bottom of the test box and seal the edges with waterproof glue to prevent the loss of small-sized sand and gravel. Also fill the soil inside the test box to a height of about 70 cm, tamped and leveled.

Secondly, cut the drainage pipes and number them sequentially. Connect the numbered circular drains to the bottom of the test box in sequence as shown in Fig. 3(a). Place the bucket on the electronic scale, pour in the water, record the water volume, calculate the required mass of calcium chloride (CaCl2) and sodium bicarbonate (NaHCO3) according to the ionic concentration, weigh out the corresponding mass of the mother liquor, pour it into the bucket evenly, and stir slowly to dissolve it fully. The prepared ionic solution is injected into the water collection tank and this operation is repeated until the requirements are met.

Thirdly, the experimental data and phenomena were recorded. When the air inside the test chamber is completely exhausted, close the drain valve, then open the drain valve in the tunnel and monitor the water pressure value of the water injection hole, and keep the leakage stable when it reaches 50 kPa. Use a measuring cylinder to collect the water discharged from the drainage port, and record the time required to collect 50 ml of water. Turn off the water pump of the water collection tank, open the water valve at the bottom of the test chamber, and wait until the ground water in the test chamber is completely discharged. Remove the drainage pipe, measure the weight of each water pipe after drying and record it, and install the water pipe to its original position after the re-recording is completed. Then, test the ion concentration of ground water in the water collection tank every 12 hours and add the appropriate substance according to the change in ion concentration.

Finally, repeat the above operation until the crystal weight is stable. And complete each set of tests and record the data according to the predetermined working conditions.

Round 2

Reviewer 3 Report

The authors have responded to the points raised by this reviewer. I suggest the paper to be accepted for publication.